# Unsupervised Multi-View Object Segmentation Using Radiance Field Propagation

**Xinhang Liu**[1]  **Jiaben Chen**[2]  **Huai Yu**[3]  **Yu-Wing Tai**[1,4]  **Chi-Keung Tang**[1]

[1]HKUST    [2]UC San Diego    [3]Wuhan University    [4]Kuaishou Technology

`https://xinhangliu.com/nerf_seg`

## Abstract

We present radiance field propagation (RFP), a novel approach to segmenting objects in 3D during reconstruction given only unlabeled multi-view images of a scene. RFP is derived from emerging neural radiance field-based techniques, which jointly encodes semantics with appearance and geometry. The core of our method is a novel propagation strategy for individual objects' radiance fields with a bidirectional photometric loss, enabling an unsupervised partitioning of a scene into salient or meaningful regions corresponding to different object instances. To better handle complex scenes with multiple objects and occlusions, we further propose an iterative expectation-maximization algorithm to refine object masks. To the best of our knowledge, RFP is one of the first unsupervised approaches for tackling 3D real scene object segmentation for neural radiance field (NeRF) without any supervision, annotations, or other cues such as 3D bounding boxes and prior knowledge of object class. Experiments demonstrate that RFP achieves feasible segmentation results that are more accurate than previous unsupervised image/scene segmentation approaches, and are comparable to existing supervised NeRF-based methods. The segmented object representations enable individual 3D object editing operations.

## 1  Introduction

Despite deep learning's remarkable success in object detection and segmentation [34, 31, 11] over the past decade, several issues remain relevant. First, modern neural network approaches require a massive amount of labeled, heavily curated data [8, 19] to achieve superior performance, or they may have difficulty dealing with problems in which labeled data are lacking. In addition, most segmentation methods regard images as flat 2D arrays of pixels where pixel groupings form object instances. Natural images are 2D projections of the underlying 3D world. Thus, discarding one of the dimensions gives rise to fundamental ambiguities and challenges to the task resulting from partial or total occlusion, especially for single-image methods.

The above fundamental issues motivate us to reconsider the problem of object detection and segmentation from a 3D perspective. In this paper, we study the problem of automatically partitioning a 3D scene into an arbitrary number of salient or meaningful regions, given only 2D images of the scene from multiple viewpoints.

Estimating semantic labels of a 3D scene is closely related to predicting its geometry and appearance. Therefore, recent success in novel view synthesis achieved by Neural Radiance Field (NeRF) [23] and its many follow-up work [24, 46, 3, 9] will likely make a long-lasting impact for semantic scene understanding. For example, SemanticNeRF [51] extends radiance fields to jointly encode semantics with appearance and geometry; its multi-task learning setting enables smooth and coherent semantics prediction. Nonetheless, as a supervised approach, SemanticNeRF requires semantic labels for full

36th Conference on Neural Information Processing Systems (NeurIPS 2022).

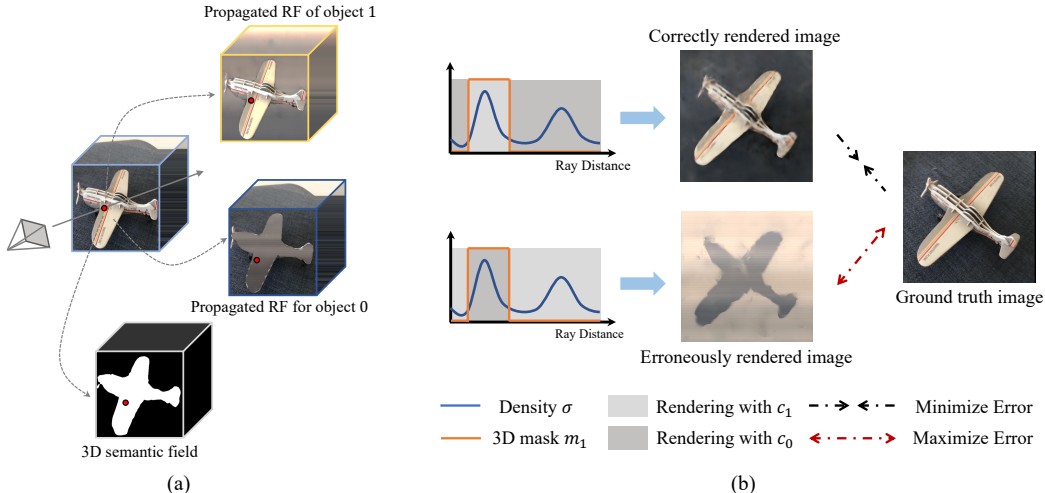

Propagated RF of object 1

Correctly rendered image

Propagated RF for object 0

Ray Distance

Ground truth image

Ray Distance

Erroneously rendered image

3D semantic field

— Density $\sigma$    Rendering with $c_1$    $-\cdot\rightarrow \leftarrow\cdot-$ Minimize Error

— 3D mask $m_1$    Rendering with $c_0$    $\leftarrow-\cdot-\cdot\rightarrow$ Maximize Error

(a)             (b)

Figure 1: Overview of *radiance field propagation (RFP)*. (a) Semantic field and propagated radiance field for object $0$ (background) and object $1$ (foreground); (b) volume rendering with correct and erroneous coloring, followed by applying the bidirectional photometric loss. The 3D semantic field is learnt in such an unsupervised manner. For clarity, we show here a single salient foreground object example.

supervision. When labels are incomplete or noisy, this approach still relies on a cross-entropy loss between the prediction and the target label to supervise its semantic branch. Previous work [48] and concurrent work [36] studied unsupervised discovery of object field to enable 3D scene segmentation and editing. But their system needs pre-training on datasets with specific object categories, such as CLEVR [14] and ShapeNet [41] objects, thus not capable of segmenting arbitrary objects in general scenes.

To explore fully self-supervised image segmentation in the multi-view setting, we study the task of partitioning a scene into multiple object instances with only multi-view images and calibrated camera poses. In this setting, annotated labels in any form are not available. Furthermore, there are no other cues, such as 3D bounding boxes for the objects or prior knowledge about object classes.

To tackle this challenging task, we propose radiance field propagation (RFP) based on the emerging radiance field-based scene representation. Fig. 1 shows an overview of our method. We use a continuous 3D semantic field like [51] to model semantic information in space. Note that we do not model object geometry and appearance by a single unified radiance field, but rather by a layered scene representation similarly adopted in [49] and [43], representing each entity (each foreground object and the background) as an independent radiance field. Based on the layered scene representation, we introduce a propagation strategy on each radiance field. The input to these continuous radiance field functions can be **(a)** a coordinate occupied by the corresponding object, or **(b)** a coordinate occupied by other objects or no object. For **(a)**, we expect the function's output to be an accurate model of the object's geometry and appearance. We propagate the statistical bias of appearance from **(a)** to **(b)**. (Note that the density function or geometry is not propagated.)

Inspired from the unsupervised object detection and segmentation approaches [45, 44, 33], we assume that an accurate partitioning of the scene should make the estimation of one part from others difficult. If we use the semantic field to distinguish **(a)** and **(b)**, then an accurate semantic field should maximize the error between the propagated appearance from **(a)** to **(b)** and the ground truth appearance. A bidirectional photometric loss term is thus proposed that maximizes the error of erroneously rendered image from propagated appearance, while minimizing the error of correctly rendered image, so that self-supervision of the learning of the semantic fields can be achieved.

While the above techniques can segment well scenes with a single salient object, to handle more complex scenes with multiple objects and occlusions, we design a post-processing procedure to refine object masks based on iterative expectation-maximization. These procedures improve the 3D segmentation of each object in scenes with multiple objects. With individually segmented object NeRFs, we can render individual objects and even enable various 3D scene editing, including object insertion, duplication, translation, and rotation, which is achieved by the previous works [49, 43] only when using manual annotation.

As the first pertinent unsupervised multi-image method, the experiments show our system performs better than the SoTA methods for unsupervised single image segmentation or 3D scene segmentation, and is comparable with supervised NeRF-based novel view semantic synthesis approaches.

In summary, the contributions of this paper are as follows.

- We are among the first to study fully unsupervised multi-view image segmentation with real scene NeRF without any annotations such as 3D bounding boxes, or prior knowledge of object class to the best of our knowledge.

- We exploit layered radiance fields and propose Radiance Field Propagation with a bidirectional photometric loss to guide the training of the semantic field.

- We design an iterative expectation-maximization algorithm to refine object masks for handling complex scenes.

- Extensive experimental validation and extensive ablation studies justify the design of each component and demonstrate the effectiveness of our system on applications such as individual object rendering and editing.

## 2 Related Work

**Unsupervised Single Image Segmentation.** Despite the success of supervised deep learning based methods [20, 32, 1, 50, 4, 5], much effort has been made on unsupervised single image segmentation. Different from supervised image segmentation, where pixel-level semantic labels such as floor or table, unsupervised image segmentation aims to predict more general labels, such as foreground and background. To solve unsupervised category-agnostic image segmentation, which could be regarded as primarily a grouping or clustering problem, as opposed to a labeling task, using color, contrast, and hand-crafted features to cluster pixels has been investigated [7, 13, 15, 17]. More recent approaches use generative models to partition an image [6, 2].

Notably, in Contextual Information Separation (CIS) [45], object detection and segmentation from an information-theoretic perspective were first studied. To detect moving objects in images, CIS uses a segmentation network to minimize the mutual information between the inside and the outside of putative motion regions. This procedure is however sensitive to motion errors. To further enable stationary objects detection and segmentation, DyStaB [44] partitions the motion field by minimizing the mutual information between segments which are then used to learn object models. While in CIS [45] inpainting is discussed on any function of the image (color histogram or optical flow), later in [33] (with somewhat similar technicalities to [45]) the authors consider inpainting on just the image itself and does not involve training a deep network with external data.

Nevertheless, all these works perform segmentation on a single image basis without considering the 3D information of the scene underlying the 2D image. In addition, the class-agnostic segmentation approaches can only tackle images with single salient object such as those from [39, 47, 26] while we consider scenes with an arbitrary number of salient objects.

**Radiance field-based scene representations.** Our unsupervised segmentation work builds on Neural Radiance Fields (NeRF) [23], which represents a scene using a multi-layer perceptron (MLP) that maps positions and directions to densities and radiances. Following work [46, 9, 3] improve NeRF for faster training and inference and more realistic rendering. Dynamic neural representations enable 4D reconstruction and rendering [29, 49, 28, 40]. Using MLP [23] or explicit feature grids [9, 3], these radiance field-based scene representations achieve unprecedented novel view synthesis effects. Our system can be built upon any radiance field-based scene representation.

**NeRF with semantics and object decompositions.** Previous works have investigated inferring semantics and achieving object-level scene understanding using NeRF [51]. In particular, Semantic-NeRF [51] adds an extra head to NeRF to predict semantic labels at any 3D position. Recent work PNF [18] can estimate a panoptic radiance field representation of any scene from just color images. However, SemanticNeRF [51] takes 2D semantic label maps as input, while PNF [18] is prior-based and trained with external annotated data. In [49, 43, 27, 10], a scene was decomposed into a set of NeRFs associated with foreground objects separated from the background. However, [49, 43] cannot produce 3D segmentation of objects to enable rendering and editing of individual objects. ST-NeRF[49] and ObjectNeRF [43] represent each entity with separated NeRFs. While enabling

rendering and editing of a single object, they require semantic information provided by the label map for supervision. GIRAFFE [25] represents scenes as compositional generative neural feature fields, enabling disentanglement of individual objects for image synthesis without studying obtaining 3D scene segmentation. [37] turns a single image of a scene into a 3D model represented as a set of NeRFs, with each of them corresponding to a different object. They enable unsupervised 3D scene segmentation and novel view synthesis, with experiments are on synthetic datasets. After that [48] and concurrent work [36] studied unsupervised discovery of object fields which enables scene segmentation and editing in 3D. But their systems are prior-based and need pre-training on datasets with specific object categories, such as CLEVR [14] and ShapeNet [41] objects, thus not capable of segmenting arbitrary objects in general scenes. Unlike all these methods, our method can segment objects in 3D scenes and perform individual object rendering and editing without any semantic annotation and knowledge of object class.

## 3 Method

With a set of posed images of a 3D scene as input, our goal is to reconstruct the scene and segment each of the object instances. We view each salient object in the foreground as an object instance, and the background an extra separate instance. We assume there are in total $K$ salient foreground object instances in the scene (we denote foreground objects as instance 1 to $K$ and use instance 0 to denote the background as instance 0 in addition).

### 3.1 Propagation of Layered Radiance Field

Extented from NeRF [23], we represent the continuous volume density, emitted color, and semantics [51] as functions with a spatial coordinate $\mathbf{x} = (x, y, z)$ and viewing direction $\mathbf{d} = (\theta, \phi)$ as input. Specifically, we represent volume density as a function that maps a world coordinate $\mathbf{x} = (x, y, z)$ to the continuous 3D scene density $\sigma$:

$$\sigma(\mathbf{x}) = F_\sigma(\mathbf{x}; \boldsymbol{\Theta}). \tag{1}$$

Following [51], we formalize semantic information as an inherently view-invariant function that maps only a world coordinate $\mathbf{x}$ to a distribution over $K + 1$ labels corresponding to $K + 1$ object instances via pre-softmax semantic logits $\mathbf{s}(\mathbf{x}) = (s_0, \ldots, s_K)$:

$$\mathbf{s}(\mathbf{x}) = F_s(\mathbf{x}; \boldsymbol{\Theta}). \tag{2}$$

Here we only consider $K + 1$ labels for all practical instances, without an additional label for non-object coordinates of which the contribution would be eliminated automatically by a value close to 0 of $F_\sigma(\mathbf{x})$. With semantic logits of each coordinate, we compute the label to assign a coordinate to by argmax over all the objects.

$$l(\mathbf{x}) = \mathbf{s}_{\texttt{argmax}}(\mathbf{x}). \tag{3}$$

Since the operation of `argmax` is not differentiable, we use the straight through trick [12, 21, 38, 42] and compute the assignment as

$$\hat{\mathbf{l}}(\mathbf{x}) = \texttt{one-hot}(\mathbf{s}_{\texttt{argmax}}(\mathbf{x})) + \mathbf{s}(\mathbf{x}) - \texttt{sg}(\mathbf{s}(\mathbf{x})), \tag{4}$$

where `sg` is the stop gradient operator. $\hat{\mathbf{l}}(\mathbf{x})$ has the one-hot value of assignment to an object, whose gradient is equal to the gradient of $\mathbf{s}(\mathbf{x})$, thus making the system differentiable and allowing gradient back propagation. We obtain the 3D mask for object $k$ as

$$m_k(\mathbf{x}) = \begin{cases} 1, & l(x) = k, \\ 0, & l(x) \neq k. \end{cases} \tag{5}$$

We further define $\overline{m} = 1 - m$ as the inversion operation on masks. Instead of a unified function to represent the continuous field of the emitted color of the whole scene, we use $K + 1$ functions to model the appearance of $K + 1$ instances. Such individual modeling of each object with a different function is similar to some previous work [49, 43]. For object $k$, we represent its color as a function that maps a world coordinate $\mathbf{x} = (x, y, z)$ as well as view direction $\mathbf{d} = (\theta, \phi)$ to the continuous 3D scene color $\mathbf{c}_f = (r, g, b)$:

$$\mathbf{c}_k(\mathbf{x}, \mathbf{d}) = F_c^k(\mathbf{x}, \mathbf{d}; \boldsymbol{\Theta}). \tag{6}$$

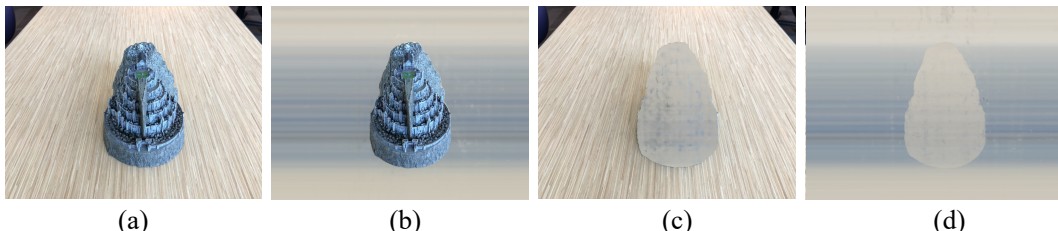

(a)          (b)          (c)          (d)

Figure 2: For clarity this example shows one salient foreground object. Object 0 and 1 respectively denote the background and the foreground object: (a) corrected rendered image, (b) image rendered with object 1's radiance field, (c) image rendered with object 0's radiance field and (d) image with object 0 rendered with object 1's RF and object 1 with object 0's RF.

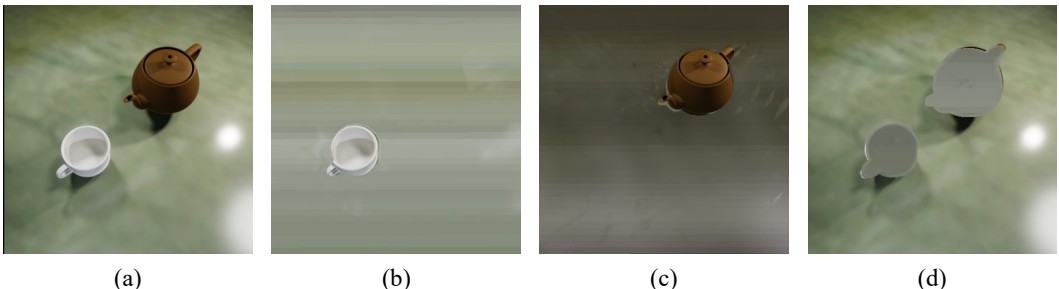

(a)          (b)          (c)          (d)

Figure 3: Illustration of Radiance Field Propagation on a multi-object scene. (a) corrected rendered image, (b) image rendered with object 1's radiance field, (c) image rendered with object 2's radiance field and (d) image rendered with object 3's (background) radiance field

Starting here we use $\mathbf{c}_k(\mathbf{x})$ instead of $\mathbf{c}_k(\mathbf{x}, \mathbf{d})$ for notation simplicity. The input world coordinate $\mathbf{x}$ to this function can be an arbitrary coordinate in the scene, i.e., either occupied by object $k$ or otherwise. The 3D object mask in Eqn. 5 naturally distinguishes these two cases. For the former case, $m_k(\mathbf{x}) = 1$, and we expect $\mathbf{c}_k(\mathbf{x})$ to be the precise color of the foreground object to achieve realistic rendering results. In the latter case, $m_k(\mathbf{x}) = 0$, and we encourage $\mathbf{c}_k(\mathbf{x})$ here to be propagated from coordinates in the former case and have a value of color close to object $k$. This 3D propagation procedure is enforced by the propagation regularizer

$$\mathcal{L}_{\text{prop}} = \sum_{k=0}^{K} \|\overline{m_k(\mathbf{x})}(\mathbf{c}_k(\mathbf{x}) - \mu)\|_2^2, \tag{7}$$

where $\mu$ is the average value of $m_k(\mathbf{x})\mathbf{c}_k(\mathbf{x})$ (not counting masked out values). The average operation is performed on the training batch. At coordinates not occupied by object $k$, the propagated value is similar to one at a coordinate that belongs to object $k$, which can be regarded as a prediction of the radiance of a partition other than object $k$ knowing object $k$'s radiance field.

### 3.2 Bidirectional Photometric Loss

Since an accurate segmentation of object $k$ should maximize the error of this appearance prediction, a loss term Eqn. 12 encouraging such maximization of error would guide the optimization of the semantic field without supervision from any ground-truth annotations.

For each coordinate, we calculate the value of all $K + 1$ color fields at it. The correct assignment for this coordinate color is given by

$$\mathbf{c}(\mathbf{x}) = \sum_{k=0}^{K} m_k(\mathbf{x})\mathbf{c}_k(\mathbf{x}), \tag{8}$$

If we artificially color object $i$ with $\mathbf{c}_j(\mathbf{x})$, and color objects other than object $i$ with the color of object $\mathbf{c}_i(\mathbf{x})$, we get a deliberately erroneous color field

$$\tilde{\mathbf{c}}_{i,j}(\mathbf{x}) = m_i(\mathbf{x})\mathbf{c}_j(\mathbf{x}) + \overline{m_i(\mathbf{x})}\mathbf{c}_i(\mathbf{x}). \tag{9}$$

To compute the color of a single pixel, with either the correct or an erroneous radiance field, we approximate volume rendering by numerical quadrature, the same way as [23]. Let $\mathbf{r}(t) = \mathbf{o} + t\mathbf{d}$

be the ray emitted from the centre of projection of camera space through a given pixel, traversing between near and far bounds ($t_n$ and $t_f$), then for selected $P$ random quadrature points $\{t_k\}_{p=1}^{P}$ between $t_n$ and $t_f$, the approximated expected color using $\mathbf{c}(\mathbf{x})$ and the erroneous color using $\tilde{\mathbf{c}}_{i,j}(\mathbf{x})$ are given by:

$$\hat{\mathbf{C}}(\mathbf{r}) = \sum_{p=1}^{P} \hat{T}(t_p)\,\alpha\,(\sigma(t_p)\delta_p)\,\mathbf{c}(t_p)\,, \tag{10}$$

$$\tilde{\mathbf{C}}_{i,j}(\mathbf{r}) = \sum_{p=1}^{P} \hat{T}(t_p)\,\alpha\,(\sigma(t_p)\delta_p)\,\tilde{\mathbf{c}}_{i,j}(t_p)\,, \tag{11}$$

where $\hat{T}(t_p) = \exp\left(-\sum_{p'=1}^{p-1}\sigma(t_p)\delta_p\right)$, $\alpha\,(x) = 1 - \exp(-x)$, and $\delta_p = t_{p+1} - t_p$ is the distance between two adjacent quadrature sample points. We show examples of images using different rendering scheme in Fig. 2 and Fig. 3. To train our system, we minimize the error of the correctly rendered colors (Fig. 2(a)), and at the same time maximize the error of the erroneously rendered colors (Fig. 2(d)). This yield to the bidirectional photometric loss function to train our system.

$$\mathcal{L}_{\text{photo}} = \sum_{\mathbf{r}\in\mathcal{R}} \left\|\hat{\mathbf{C}}(\mathbf{r}) - \mathbf{C}(\mathbf{r})\right\|_2^2 - \frac{1}{K(K+1)} \sum_{i=0}^{K} \sum_{j\in\{0,\ldots,K\}\setminus\{i\}} \sum_{\mathbf{r}\in\mathcal{R}} \left\|\tilde{\mathbf{C}}_{i,j}(\mathbf{r}) - \mathbf{C}(\mathbf{r})\right\|_2^2, \tag{12}$$

where $\mathcal{R}$ are the sampled rays within a training batch. We compute the expected semantic logits of a single pixel in an image the same way as [51]:

$$\hat{\mathbf{S}}(\mathbf{r}) = \sum_{p=1}^{P} \hat{T}(t_p)\,\alpha\,(\sigma(t_p)\delta_p)\,\mathbf{s}(t_p)\,, \tag{13}$$

which can then be transformed into multi-class probabilities through a softmax normalization layer. The 2D label map and object mask for object $k$ are

$$L(\mathbf{r}) = \hat{\mathbf{S}}_{\text{argmax}}(\mathbf{r})\,, \tag{14}$$

$$M_k(\mathbf{r}) = \begin{cases} 1, & L(\mathbf{r}) = k, \\ 0, & L(\mathbf{r}) \neq k. \end{cases} \tag{15}$$

We also found that estimating a coarse 2D label map for each input image using an unsupervised single image-based approach and then computing a cross-entropy loss between the map and the estimated semantic logits can improve the results of our system:

$$\mathcal{L}_{\text{init}} = -\sum_{\mathbf{r}\in\mathcal{R}} \sum_{k=0}^{K} S_{\text{init}}^{k}(\mathbf{r}) \log \hat{S}^{k}(\mathbf{r}), \tag{16}$$

where $\mathbf{S}_{\text{init}}$ is the initial coarse label map. Since the approach we use is unsupervised and does not involve external data, so this initial stage does not undermine this property of our method. Hence the total training loss $L$ combining the bidirectional photometric loss Eqn. 12, the propagation regularizer Eqn. 7 and the initial semantic estimation loss Eqn. 16 is:

$$\mathcal{L} = \mathcal{L}_{\text{photo}} + \lambda_{\text{prop}}\mathcal{L}_{\text{prop}} + \lambda_{\text{init}}\mathcal{L}_{\text{init}}\,, \tag{17}$$

where $\lambda_{\text{prop}}$ and $\lambda_{\text{init}}$ are weights of corresponding loss terms.

### 3.3 EM-based Mask Refinement for Complex Scenes

In the previous subsection we have described the core components necessary for performing 3D segmentation, and our experiments have shown the effectiveness of the proposed system on segmenting scenes with one salient object. To extend the segmentation on more complex scenes with multiple foreground objects and occlusion among them, we further propose an EM algorithm-based procedure to refine the object mask as post-processing.

For each input image, we first feed it into a deep feature extractor, and cluster all the pixels into two classes using $k$-means clustering. In this way we get initial 2D image masks for the foreground (consisting of $K$ objects) $S_{\text{foreground}}$ and the background $S_{\text{background}}$. We apply differentiable

feature clustering (DFC) [17] to further partition foreground masked by $S_{\text{foreground}}$ into $K$ parts, $\left\{S_{\text{init}}^k\right\}_{k=1,\ldots,K}$ and let $S_{\text{init}}^0 = S_{\text{background}}$, for computing the initial semantic estimation loss Eqn. 16. Note that here we introduce a deep feature extractor [35] trained on large-scale data, which is the only introduction of external data to our method. But since no annotated data is involved, the unsupervised nature of our method is not undermined. DFC results in a response vector map $\mathbf{v}(x)$ for each image, where the $x$ is a pixel index. Pixels belonging to the same object tend to have similar response vectors. After the training of our system, expectation-maximization (EM) is employed to refine the output mask obtained from Eqn. 15 with these response vector maps. Let $\boldsymbol{\mu}_k$ be a mean vector of object $k$, and we initialize $\boldsymbol{\mu}_k$ as

$$\boldsymbol{\mu}_k^{(0)} = \frac{\sum_x M_k(x)\mathbf{v}(x)}{\sum_x M_k(x)}. \tag{18}$$

We formulate the posterior probability of $\mathbf{v}$ given $\boldsymbol{\mu}_k$ and the latent variable $z_k$ at $t$-th EM iteration as

$$p(\mathbf{v}|\boldsymbol{\mu}_k) = \exp(-||\mathbf{v} - \boldsymbol{\mu}_k||_2^2), \tag{19}$$

$$z_k^{(t)}(x) = \frac{p(\mathbf{v}(x)|\boldsymbol{\mu}_k^{(t)})}{\sum_{l=0}^K p(\mathbf{v}(x)|\boldsymbol{\mu}_l^{(t)})}. \tag{20}$$

At each EM iteration, we update the $\boldsymbol{\mu}$ as

$$\boldsymbol{\mu}_k^{(t+1)} = \frac{\sum_x z_k^{(t)}(x)\mathbf{v}(x)}{\sum_x z_k^{(t)}(x)} \tag{21}$$

The final semantic logits are given a weighted sum of $\mathbf{Z}$ concatenated from $\left\{z_k^T\right\}_{k=0,\ldots,K}$ at $T$-th iteration and estimated logits after training in Eqn. 13:

$$\mathbf{S}_{\text{final}} = \mathbf{Z} + w\hat{\mathbf{S}}. \tag{22}$$

where $w$ is a weight. Therefore, the refined mask can be computed similarly to Eqn. 14 and Eqn. 15.

With scene segmentation obtained using our system, applications such as single object rendering and editing can be achieved. **We defer the description of the relevant procedure in the supplemental material.**

## 4 Experiments

We evaluate our method on three datasets with different complexities. We report qualitative and quantitative results on scene segmentation with comparison to previous work. We also show the application of our method on individual object rendering and editing. We conduct ablation studies to justify each component of our method. **We gently urge readers to check the supplementary material for more qualitative results in the form of pictures and videos, as well as the settings of our experiments in detail.**

**Datasets.** We first test RFP on scenes with a single foreground object using two real datasets, **Local Light Field Fusion (LLFF)** [22] and **Common Objects in 3D (CO3D)** [30]. As LLFF does not provide ground-truth label maps, we only show qualitative results on the LLFF dataset. To test RFP on scenes with multiple objects, we build a synthetic dataset upon ClevrTex [16]. We split all the datasets into training views and testing views with a ratio of around 9 to 1.

**Metrics.** We adopt the widely-used pixel classification accuracy (Acc.) and mean intersection over union (mIoU) as our metric. To evaluate scene segmentation in 3D, we report the metrics computed for both training and novel views. The former enables direct comparison to 2D methods, and the latter reflects the 3D nature. We denote Acc. and mIoU of novel views as N-Acc. and N-mIoU. Note that for our method, the difference between the training and the novel view is only related to whether the color image of the view is provided. The semantic label of any view is not input. Differently, for the supervised methods [51, 43], both the color image of the training view and the ground truth semantic label are provided as inputs.

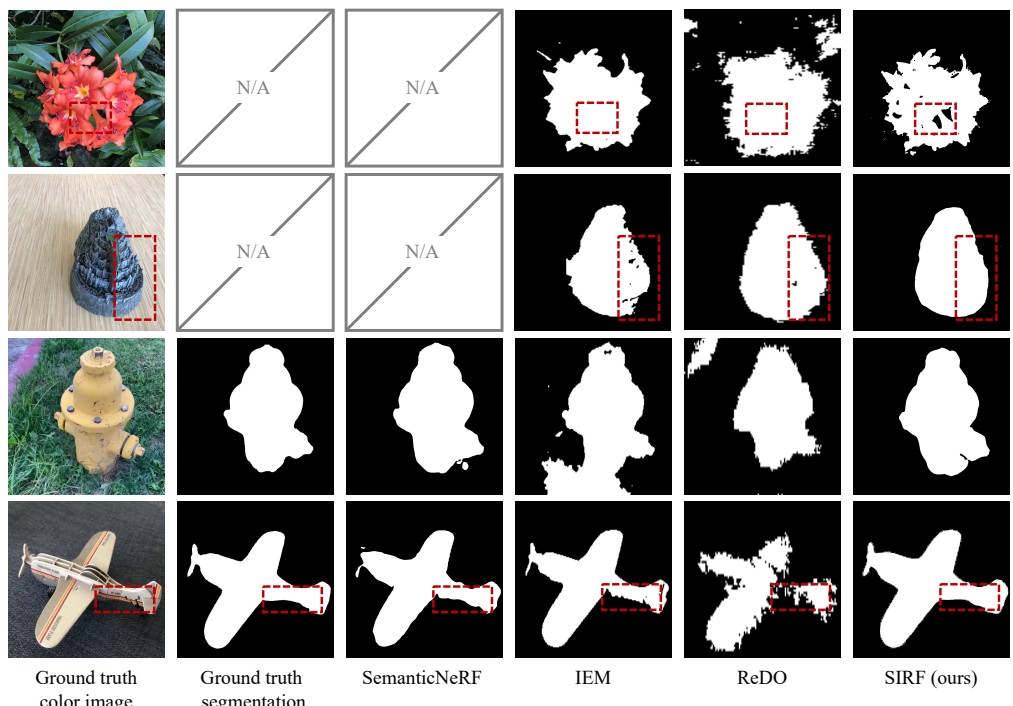

| Ground truth color image | Ground truth segmentation | SemanticNeRF | IEM | ReDO | SIRF (ours) |

Figure 4: Qualitative comparison on 3D segmentation on scenes with a single foreground object. IEM [33] and ReDO [6] are unsupervised single image-based methods. There are no ground truth labels for the LLFF dataset [22], and thus the supervised approach SemanticNeRF [51] is not applicable. Images are cropped for typography.

| Methods | CO3D [30] | | | | RFP Synthetic | | | |
|---|---|---|---|---|---|---|---|---|
| | Acc. ↑ | mIoU ↑ | N-Acc. ↑ | N-mIoU ↑ | Acc. ↑ | mIoU ↑ | N-Acc. ↑ | N-mIoU ↑ |
| ReDO [6] | 86.7 | 63.9 | - | - | - | - | - | - |
| IEM [33] | 95.8 | 83.4 | - | - | - | - | - | - |
| DFC [17] | - | - | - | - | 95.7 | 79.4 | - | - |
| uORF [48] | - | - | - | - | 91.7 | 32.6 | 92.1 | 27.6 |
| SemanticNeRF [51] | 99.0 | 96.0 | 98.7 | 95.1 | 99.6 | 96.6 | 99.3 | 94.8 |
| ObjectNeRF [43] | - | - | - | - | 99.6 | 96.6 | 99.2 | 94.5 |
| RFP (W/O PROPAGATION) | 96.2 | 84.5 | 95.4 | 57.7 | 84.6 | 85.3 | 96.3 | 85.1 |
| RFP (W/O INIT. EST.) | 76.3 | 49.4 | 73.5 | 49.0 | 93.7 | 60.8 | 94.1 | 55.3 |
| RFP (W/O EM REFINEMENT) | - | - | - | - | 96.8 | 86.5 | 97.1 | 86.9 |
| RFP (ours) | **98.5** | **94.1** | **97.8** | **94.0** | **97.7** | **88.0** | **97.4** | **87.5** |

Table 1: Quantitative comparison and ablation. N-Acc. and N-mIoU are Acc. and mIoU evaluated on novel views. The best results without supervision are boldfaced.

## 4.1 3D Segmentation of Scenes with a Single Foreground Object

Because there is no previous work with the same unsupervised multi-view setting as our work, we compare with SoTA unsupervised class-agnostic segmentation methods, ReDO [6] and IEM [33], which ensembles a grouping or clustering problem with single images without considering 3D nature. We also compare RFP with supervised novel view semantic synthesis method SemanticNeRF [51].

We show quantitative results in Tab. 1 and qualitative results in Fig. 4. Quantitatively, RFP outperforms all single 2D image-based methods on all metrics. Qualitatively, compared to single 2D image-based methods [33, 6], our method's results have sharper and smoother edges, with no holes in the interior of objects due to the 3D nature of the scene. Our results are even comparable to the supervised method SemanticNeRF [51] on the CO3D dataset [30], which may suffer from the noisy ground truth masks of the CO3D.

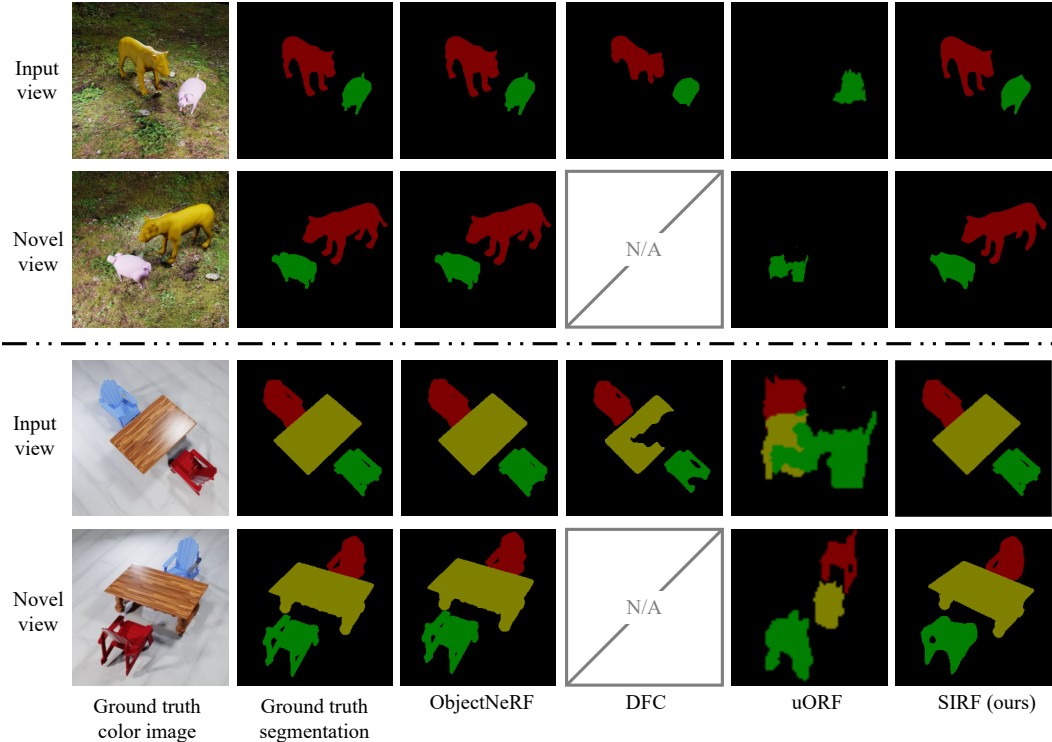

Figure 5: Qualitative comparison on multi-object scene segmentation in 3D on RFP synthetic dataset. Novel view color images are for reference and are not input. Images are cropped to fit typography.

## 4.2 3D Segmentation of Scenes with Multiple Objects

When tackling scenes with multiple objects and occlusion, we further perform the EM-based refinement introduced in Sec. 3.3. While using a deep feature extractor trained on large-scale data means introducing external data to our system here for the only time, since no annotated data is involved, the unsupervised nature of our method is not undermined.

We compare RFP with the unsupervised segmentation method DFC [17]. Since DFC relies on an initial separation of the foreground and the background, we perform $k$-means clustering beforehand. Such $k$-means clustering followed by DFC segmentation is also the pre-processing procedure mentioned in Sec. 3.3. We also compare RFP with unsupervised object radiance field discovery (uORF) [48], as well as supervised NeRF-based methods: ObjectNeRF [43] and SemanticNeRF [51].

We show quantitative results in Tab. 1 and qualitative results in Fig. 5. Quantitatively, RFP outperforms all unsupervised image/scene segmentation methods on all metrics. Qualitatively, single 2D image-based methods [17] miss more valid detection than ours. uORF can hardly give good results, probably because the method was pre-trained on a dataset consisting of naive scenes and thus with a large domain gap with our evaluation data.

## 4.3 Individual Object Rendering and Scene Editing

Following the procedure detailed in the **supplemental material**, our approach achieves depth-aware and photo-realistic free-viewpoint single object rendering and editing results. Fig. 6 and **supplemental matierials** show examples of such applications. Fig. 6 (a) & (d) are color images rendered from novel views. Fig. 6 (b) & (e) show single object rendering. Fig. 6 (c) shows the editing effect of duplicating and repositioning of the duplicated blue chairs. Fig. 6 (f) shows the editing effect of rotating the helicopter.

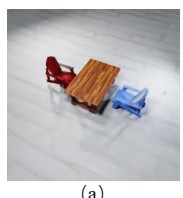 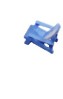 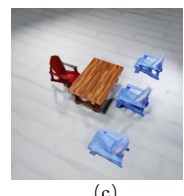 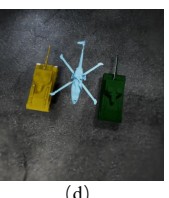 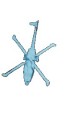 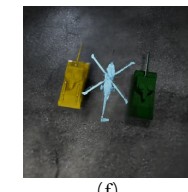

(a) (b) (c) (d) (e) (f)

Figure 6: Examples of individual object rendering and editing as an application of our approach. (a) & (d) rendered novel view of the scenes. (b) & (e) single object rendering. (c) & (f) object editing operations such as duplication, translation and rotation.

## 4.4 Ablation Study

Here, we evaluate the performance of different components and loss terms in our approach. Let W/O PROPAGATION, W/O INIT. EST. and W/O EM REFINEMENT denote the respective variations of our approach without the radiance field propagation and using vanilla photometric loss instead of the bidirectional loss, without the initial semantic estimation loss, and without the EM refinement. As shown in Tab. 1 quantitatively and in the supplementary material qualitatively, our complete approach achieves more accurate results than all the above variations. The radiance field propagation and the bidirectional photometric loss ensure good utilization of the 3D nature of scenes, which helps to avoid ragged edges, holes in the object's interior, and false-positive estimates outside of the object. Although unsupervised single image segmentation approaches are shown to be not robust in most of the cases, it is still essential for our system as a valid initialization. The EM-based refinement further brings good edges when tackling complex scenes with multiple objects.

## 5 Limitation

As the first approach dealing with unsupervised multi-view segmentation, our method may have several limitations. To begin with, our method may have difficulty when the object of interest is severely occluded, textured or with similar appearance to the background. In addition, our work mainly takes advantage of appearance instead geometric cues. Thus using generative method to perform geometry inpainting may lead to a fruitful direction for future exploration. Third, the dependence on a pretrained feature extractor should be alleviated.

## 6 Conclusion

In this work, we present one of the first approaches in real scene NeRF for tackling unsupervised multi-view image segmentation using radiance field propagation with a bidirectional photometric loss to guide the reconstruction of semantic field. We design an EM-based mask refinement procedure to handle complex scenes. Our experiments show the effectiveness of our approach, with useful applications on NeRF such as individual object rendering and editing.

## Acknowledgements

We would like to thank Yichen Liu and Shengnan Liang for fruitful discussion at the inception stage of the project. This research is supported in part by the Research Grant Council of the Hong Kong SAR under grant no. 16201420.

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
