# OpenReview forum: "Unsupervised Multi-View Object Segmentation Using Radiance Field Propagation"
_NeurIPS.cc/2022/Conference — NeurIPS 2022 Accept_

### Official Review · Reviewer_QA4f · 2022-07-10

**Rating:** 6
**Confidence:** 4
**Soundness:** 3 good
**Presentation:** 3 good
**Contribution:** 3 good

**Summary:**

This paper focuses on unsupervised segmentation in 3D with multiview images. The main idea is to use a mixture model of multiple visually contrastive radiance fields to compositionally reconstruct the scene. The contrastiveness among the RFs is enforced by propagating average appearance to empty space and maximizing the visual differences between the reconstruction and the inversely mixed composition. Experiments show good results on a single-object real dataset and a multi-object synthetic dataset.

**Questions:**

It would be interesting to see if the same method works with single images. Have you ever tried that with 2D or 3D RFs?

**Limitations:**

Yes

**Strengths And Weaknesses:**

Strength:

+ Interesting idea to create and use contrastiveness in compositional NeRFs for unsupervised segmentation.
+ In general, the experimental results look good.
+ Experiments on a real dataset.
+ Comprehensive comparison to the state of the art.

Weakness:

- I feel a few aspects in technical description and experiments are not well clarified and justified. For Eq. (4), why does it need to be differentiable? It seems l(x) is only used in Eq. (5), but is discontinuous and not differentiable. Why are numbers in 2nd last row of Table 1 missing?

---

> ### Author Response · Authors · 2022-08-01
> **Response to Reviewer QA4f**
>
> Thank you for your positive feedback as well as thoughtful suggestions and questions. Below, we address your points individually.
>
> ### Explanation for Eqn. 4 and 5 (Weakness #1)
> Please check the explanation in the general response. Specifically, differentiability is necessary for backpropagation of the relevant gradient to the pre-softmax semantic logits $\mathbf{s}(\mathbf{x})$.
>
> ### 2nd last row of Tab. 1 (Weakness #2)
> The EM refinement is designed for scenes with multiple objects. We found that without this refinement, the results on single-object scenes like those from CO3D or LLFF are satisfactory. Our full model for single object scenes does not contain EM refinement.  The EM refinement involves a strong feature extractor and we thought this might be overkill for single-object scenes. Involving such a feature extractor pretrained on large-scale data would also cause confusion when compared with unsupervised single image segmentation methods IEM and ReDO. We report a quantitative ablation on performing EM refinement on single object scenes in the following table.
>
> |              | Acc. $\uparrow$ | mIoU  $\uparrow$ | N-Acc. $\uparrow$ |N-mIoU  $\uparrow$ |
> |  ----------------|-------------|------------------|---------------------- |-----------------|
> |w/o EM refinement |  97.9   |   94.1  |   97.4  |       93.6 |
> |w/ EM refinement |98.0    |    94.1     |  97.4         |93.6    |
>
>
>
> ### Trying our method on single images (Question)
> Thanks for this suggestion and such results would be insightful for the readers. We try our method with only one training view (input single image at a time) as input with qualitative results available here: [anonymous link](https://anonymous.4open.science/r/NeurIPS-Rebuttal-Materials-PaperID3608/single%20image.pdf). Quantitative metrics are reported in the following table. We also report qualitative metrics on the same scene with all 102 images as input at a time for comparison.
>
> | # of input views|Acc. $\uparrow$|mIoU  $\uparrow$|
> |  ----------------|-------------|------------------|
> |1  |  87.8     |  68.1   |
> |102  |  97.1  |   92.8      |
>
>
> On the other hand, our paper's notable contribution exactly consists of handling 3D RFs, so we are a little unclear why you mention 2D or 3D RFs in the questions. But we are willing to discuss this further in the Reviewer-Author Discussions session.

---

> > ### Comment · Reviewer_QA4f · 2022-08-07
> > **Thanks for response**
> >
> > Thank the authors for responses. I stand by acceptance.

---

> > > ### Author Response · Authors · 2022-08-09
> > > **Thanks for the feedbacks**
> > >
> > > Dear Reviewer QA4f,
> > >
> > > Thanks for your positive and valuable feedback, which would help us improve this work.
> > >
> > > Anonymous authors

---

> ### Author Response · Authors · 2022-08-07
> **Further discussion with Reviewer QA4f**
>
> Dear Reviewer QA4f,
>
> We hope that you had a chance to read the rebuttal (and also the general response above) as the discussion period is ending soon. In particular, we explained Eqn. 4&5, hoping it would be helpful. We responded to the question about the missing numbers and provided a quantitative ablation on performing EM refinement on single object scenes. We appreciate your idea of trying our method on single images, and we provided qualitative and quantitative results, which we hope would be insightful.
>
> Please let us know whether you have any further concerns or suggestions to improve this work’s quality.
>
> Thank you!

---

### Official Review · Reviewer_1Do1 · 2022-07-11

**Rating:** 6
**Confidence:** 3
**Soundness:** 2 fair
**Presentation:** 3 good
**Contribution:** 2 fair

**Summary:**

This paper presents an approach to learn 3D object instance segmentation, given RGB images and camera poses. The main novelty is the "propagated" radiance field obtained by summing radiance field layers weighted by *inverted* masks. Correct labeling would maximize its photometric error. The network is trained using both traditional photometric loss and the proposed loss. EM-refinement is used to improve the result.

**Questions:**

Table 1 is confusing. It says “The best results without supervision are boldfaced.” But it is not clear on its own which rows they are. If I understand correctly, ablated results perform worse, as expected. And all the other papers had some label or bounding box supervision.  It might be helpful to indicate which additional supervisory signals they used, in each row.

Was this baseline considered: train vanilla NeRF, do 3D reconstruction (e.g. marching cubes), treat each connected volumetric component as an instance (assume known floor level). It would also be unsupervised.

L157, is m_k(x) = 0 supposed to be m_k(x) = 1 ?

**Limitations:**

Yes

**Strengths And Weaknesses:**

Strength: Unlike prior work, instance segmentation is possible. And inverting the mask is an interesting idea.

Weakness: The scenes do not seem complex enough for the task. An object close to a wall (in addition to the floor) or another object may be difficult to segment. It would be insightful to see some failure cases (either real or synthetic).

---

> ### Author Response · Authors · 2022-08-01
> **Response to Reviewer 1Do1**
>
> Thanks for the thoughtful review, and the helpful suggestions. We will improve the paper based on them. For the proposed weaknesses and questions:
>
> ### Scene complexity (Weakness)
> Thank you for pointing out the weakness of our work on scene complexity. We have stressed it in the revised submission.
> We give two failure cases in this [anonymous link](https://anonymous.4open.science/r/NeurIPS-Rebuttal-Materials-PaperID3608/failure%20cases.pdf). In one of them, the object is close to a patterned wall.
> We also provide [additional results](https://anonymous.4open.science/r/NeurIPS-Rebuttal-Materials-PaperID3608/DTU%20results.pdf) on DTU dataset where the object is textured.
> Please also check the discussion on *More complex scenes and failure cases* in the general response.
>
> ### Confusion in Tab. 1 (Question \#1)
> SemanticNeRF and ObjectNeRF are two supervised methods since they directly use annotated labels or masks for supervision. All other rows are unsupervised. DFC involves a feature extractor pretrained on large-scale datasets but no labels for supervision. IEM, ReDO, and uORF are unsupervised approaches. Among them, uORF is trained on a large-scale dataset consisting of very simple scenes, but without supervision from labels. Thank you for your suggestion of adding an indication, which will further improve our paper's clarity.
>
> ### Extra baseline (Question \#2)
> This baseline is reasonable and we worked out such an implementation: We train a vanilla NeRF for the scene and extract an explicit voxel grid-based representation from it. Treating each connected volumetric component as an instance, we use 3D floodfill to get object segmentation, with manually assigned seed points.
>
> It turns out this baseline performed poorly, probably because, in most NeRF scenarios, all objects of interest tend to have some connected volumetric components.
> The quantitative result is reported in the following table and the qualitative result can be found in this [anonymous link](https://anonymous.4open.science/r/NeurIPS-Rebuttal-Materials-PaperID3608/vanilla%20NeRF%20baseline.pdf).
>
>
> |            |Acc. $\uparrow$ | mIoU  $\uparrow$ | N-Acc. $\uparrow$| N-mIoU  $\uparrow$|
> |---|---|---|---|---|
> |vanilla NeRF |   75.7    | 47.7| 79.0      |       48.8   |
> |ours  | 97.1     |     92.8       |    96.7        | 92.1  |
>
> ### Typo in L157  (Question \#3)
> Thank you for pointing out our typos and errors. We have fixed the typo in the equation in the revised submission.

---

> ### Author Response · Authors · 2022-08-07
> **Further discussion with Reviewer 1Do1**
>
> Dear Reviewer 1Do1,
>
> We hope that you had a chance to read the rebuttal (and also the general response above) as the discussion period is ending soon. We have provided more results and failure cases for the weakness you proposed, which we hope would be insightful. We also highlight our limitations on scene complexity in the revised submission. Finally, for your confusion about Tab. 1, we have clarified the supervisory signals used by compared methods. Thank you for proposing another baseline (vanilla NeRF); we adopted it and implemented it with quantitative and qualitative results.
>
> Please let us know whether you have any further concerns or suggestions to improve this work’s quality.
>
> Thank you!

---

### Official Review · Reviewer_fm5S · 2022-07-13

**Rating:** 5
**Confidence:** 3
**Soundness:** 3 good
**Presentation:** 3 good
**Contribution:** 2 fair

**Summary:**

The paper views image segmentation task from a 3D perspective, applies NeRF to the unsupervised multi-view image segmentation task for the first time. The paper guides the reconstruction of semantic field through the radiation field propagation, and proposes a mask refine procedure based on EM to handle complex scenes with multiple objects.

**Questions:**

The authors should explain why Propagation works intuitively?

**Ethics Review Area:**

["I don’t know"]

**Limitations:**

When dealing with multi-object complex scenes, this method needs a strong feature extractor, which requires large-scale data for pretraining，It's hard to tell whether it's the strong feature extractor works or the paper‘s method works. Especially for the comparasion with other methods, the introduction of this large-scale external data will make the comparison unfair.

**Strengths And Weaknesses:**

Pos:
The method is novel. NeRF is appropriatly applied to deal with multi view image segmentation tasks from the perspective of 3D, and a bidirectional photometric loss is proposed to construct the unsupervised task. The experimental improvements are remarkable.
The overall writing of the paper is smooth and the overall structure is clear.

Neg:
Lack of analysis on why propagation is useful makes it difficult to understand why it works.
When dealing with multi-object complex scenes, this method needs a strong feature extractor, which requires large-scale data for pretraining，It's hard to tell whether it's the strong feature extractor works or the paper‘s method works. Especially for the comparasion with other methods, the introduction of this large-scale external data will make the comparison unfair.

---

> ### Author Response · Authors · 2022-08-01
> **Response to Reviewer fm5S**
>
> Thank you for your time and valuable feedback. While being encouraged by your appreciation for our novelty and results, the issues you point out are crucial.
>
> ### Why does propagation work intuitively? (Question and Weakness #1)
>
> Propagation from one part or, say, one object $A$, of a 3D scene to others can be regarded as an estimation of others based on $A$. (We do so by averaging the color of $A$.)
> If we can accurately segment $A$, then the segmentation of $A$ we get should only contain the information of $A$ without including other objects or parts of the scene.
> Accurate segmentation of $A$ contains no information about objects other than $A$. Attempting to make an estimation of other objects based on A would be lousy, disagreeing, and contrasting to the real occupying object. This is our motivation for designing propagation and bi-directional photometric loss.
>
>
> ### Feature extractor (Weakness #2 and Limitation)
>
> The EM refinement, which involves such a strong feature extractor, is crucial when dealing with multi-object scenes.
> Despite that, our proposed radiance field propagation and bidirectional photometric loss are still essential, since without them we cannot even perform the segmentation.  Without the extractor, we can still get somewhat reasonable results as shown in the ablation.
> It should be reminded that the comparison between our method and DFC or uORF is fair, since DFC also involves such an extractor while uORF is trained on an external large-scale dataset. SemanticNeRF and ObjectNeRF do not need an extractor pretrained on a large-scale dataset, while they directly use annotated labels or masks for supervision which is not required by our method.

---

> ### Author Response · Authors · 2022-08-07
> **Further discussion with Reviewer fm5S**
>
> Dear Reviewer fm5S,
>
> We hope that you had a chance to read the rebuttal (and also the general response above) as the discussion period is ending soon. We appreciate your questions about the motivation for propagation and the strong feature extractor. We have given explanations for them, and we are looking forward to having further discussions with you. We also provided the revised version of our submission and further supplemental materials in the general responses, and we hope they will be helpful.
>
> Please let us know whether you have any further concerns or suggestions to improve this work’s quality.
>
> Thank you!

---

### Official Review · Reviewer_GXm6 · 2022-07-15

**Rating:** 6
**Confidence:** 4
**Soundness:** 3 good
**Presentation:** 3 good
**Contribution:** 3 good

**Summary:**

This paper propose an unsupervised multi-view object segmentation method leveraging on neural radiance field. The key motivation is that a good segmentation should  maximise the dis-similarity between content within each individual mask. Correspondingly, three key modules including radiance propagation, bi-directional photometric loss and EM-based post-processing are adopted to decompose a scene into different regions/objects. This paper presents promising qualitative results on object-centric scenes and synthetic multi-objects scenes under multiple observations.

**Questions:**


-How does the system work on objects with rich textures or similar appearance to backgrounds? For example, BlenderMVS or DTU datasets?
In addition, under this situation, will the  propagation mechanism using average colour be still helpful?

-How does the initial masks involve during training and how to balance different losses.

-Is the system sensitive to pose registration accuracy?

**Limitations:**

The authors have addressed the limitations and potential negative societal impact.

**Strengths And Weaknesses:**

Strengths

1. The paper is well motivated and the pipeline design follows the idea of making prediction  of one part from others difficult.

2. The evaluation and ablations are adequate on single-object scenes and multi-object scenes(synthetic)scenes. As far as I can tell this is the first multi-object segmentation framework in a purely unsupervised manner on complex scenes.

3. Though there are some missing details (mentioned in the weakness part), the overall writing is good.



Weaknesses


1. It is not clear to me what equation (4) means and helps the optimisation? In addition, \hat{l} is not mentioned in the remaining text.

2. The object-level NeRF is driven to predict average colour within masks across the batch, the design choice seems a bit ad-hoc.
What about predicting other related values to object regions like the maximum or minimum or other strategies?
As to objects with complex textures or various colours, I assume even the average colour may also not well represent the object appearance. Objects within the paper owns similar textures or colours, and I am not sure if the propagation is still feasible to other cases.


3. Similar to previous points, the main concerns comes from the question, how much does the system rely on appearance instead of others like geometric cues. The objects within several experiment have a relatively clear difference in colour compared to backgrounds, which makes it a good "colour segmenter", limiting the pipeline towards complex scenes.


4. From quantitative evaluations, the initial mask acts as an important role. Will the quality of the initial masks severely affect the final performance?  During training, will the initial segmentation loss be turned-off in the later phase of training to achieve better details?

5. As an unsupervised method, how to properly choose the loss weighting in Eq.17?

6. It would be good to demonstrate some illustrations about how each individual object-nerf works in and outside the object regions like Fig2, but for more complex scenes similar to Fig4 and Fig5.

7. Is the system performance sensitive to errors within camera poses?

8. Minor Issues.
- L157. m_k(x)=0 --> m_k(x)=1
- References should be added in L189 about the unsupervised single image-based approach, which is revealed late in the paper.
- L207. I suggest adding references or some extra explanations here. Otherwise it is hard for readers to immediately know what is exactly thee deep feature extractor here and how it works.

---

> ### Author Response · Authors · 2022-08-01
> **Response to Reviewer GXm6**
>
> Thank you for your time and detailed review.  We agree with your concerns and address them briefly in turn below.
>
> ### Explanation for Eqn. 4 and 5 (Weakness #1)
> Please check the explanation in the general response. Specifically, Eqn. 4 is for computing the label assignment at an arbitrary coordinate.  $\hat{\mathbf{l}}({\mathbf{x}})$ in Eqn. 5 is an alternative of $l({\mathbf{x}})$ with differentiability.
>
> ### Propagation mechanism (Weakness \#2)
> Predicting other related values to object regions like the maximum or minimum than averaging color sounds plausible. We tested such strategies and found that they gave rise to NaNs loss during training. This is probably due to extreme radiance values predicted by MLPs, which are common for NeRF.
>
> ### Performance on more complex cases (Weakness #2, 3 and Question #1)
> Please check the discussion on *More complex scenes and failure cases* in the general response. We provide qualitative results on DTU dataset where the object is textured in this [anonymous link](https://anonymous.4open.science/r/NeurIPS-Rebuttal-Materials-PaperID3608/DTU%20results.pdf).
>
> As the first approach dealing with unsupervised multi-view segmentation, our work has difficulty handling objects severely occluded or with a similar appearance to backgrounds. We have highlighted this limitation in the revised submission and provided more failure cases in this [anonymous link](https://anonymous.4open.science/r/NeurIPS-Rebuttal-Materials-PaperID3608/failure%20cases.pdf),
>
> ### Reliance on appearance instead of others like geometry (Weakness \#3)
> Our current propagation mechanism is appearance based. We have tried 3D GAN-based approaches to propagate geometry but have so far not gotten satisfactory results. This may be the direction for future exploration.
>
> ### Initial masks (Weakness \#4 and Question \#2)
> Initial masks are expected to give a rough object boundary, which will determine the approximate region of the segmented object in 3D space. This is especially important in multi-object cases so poor initialization may take its toll on the final performance. Notwithstanding, empirically, in the failure case we have given, the quality of the initial mask is actually often quite poor while the results still degrade quite gracefully.
>
> We did an ablation on the loss be turned off at 15000 iterations (out of 30000), with the results tabulated in the following table. We did not see the expected improvement in performance.
>
> | |Acc. $\uparrow$ |mIoU  $\uparrow$ | N-Acc. $\uparrow$ | N-mIoU  $\uparrow$ |
> |---|---|---|---|---|
> |w/o turning off |   97.9   |    94.1   |  97.4        |         93.6 |
> |w/ turning off  |97.0      |     92.3       |    97.0      |   93.4      |
>
>
> ### Balance losses (Weakness \#5 and Question \#2)
> We did not find a general strategy to balance the different losses in all scenes. Empirically, as long as the term $\mathcal L_{prop}$ exists, it will work, and the weight of this term $\lambda_{prop}$ is not critical.
> Setting a very small $\lambda_{init}$ would make the initialization of semantic fields hard, while setting it too large would cause a loss of details.
>
>
> ### More illustration (Weakness \#6)
> Thank you for your suggestion. An additional illustration of a multi-object case will make it easier for readers to understand. We have made such an illustration and please check it through this [anonymous link](https://anonymous.4open.science/r/NeurIPS-Rebuttal-Materials-PaperID3608/multi-object%20illustration.pdf). We also added this figure in the revised submission.
>
> ### Sensitivity to camera pose (Weakness \#7 and Question \#3)
>
> Accurate camera poses will definitely benefit the quality of segmentation, as such it is expected to yield better novel view synthesis. On the other hand, our method should be robust to camera pose inaccuracies. Compared with the LLFF dataset, the camera poses of the scenes in CO3D are not 100\% accurate, while both qualitative and quantitative results show that our method achieves reasonable segmentation in these cases.
>
> ### Minor Issues. (Weakness \#8)
> Thank you for pointing out our typos and errors. We have fixed the typo in the equation and added the suggested references in the revised submission.

---

> > ### Comment · Reviewer_GXm6 · 2022-08-08
> > **Reviewer Reponse After Rebuttal**
> >
> > Thank authors for the detailed response to my questions and concerns with additional clarifications and visualisations.
> >
> > I also appreciate the frankness of authors concerning the heavy reliance on appearance which make propsoed method struggle on relatively complex real-world scenes. I personally like the motivation and application of maximising dis-similarity of segments, especially its transferring to neural fields, though the tested scenes are a bit limited in my point of view.
> >
> > After reading all the reviews and authors' feedbacks (the single image test looks interesting), I would like to keep my positive rate towards this paper and recommend an acceptance.

---

> > > ### Author Response · Authors · 2022-08-09
> > > **Thanks for the feedbacks**
> > >
> > > Dear Reviewer GXm6,
> > >
> > > Thank you for your response and appreciation of our approach. Please rest assured that we will stress our limitations in our final version.
> > >
> > > Anonymous authors

---

> ### Author Response · Authors · 2022-08-07
> **Further discussion with Reviewer GXm6**
>
> Dear Reviewer GXm6,
>
> We hope that you had a chance to read the rebuttal (and also the general response above) as the discussion period is ending soon. We have responded to your questions and weaknesses. In particular, we provided more results and failure cases on objects with textures or similar appearance to backgrounds. We also did ablation studies on propagation mechanism and turning off $\mathcal L_{init}$. We also uploaded a revised version of our submission, with the figure for illustration as you suggested, highlighted limitations, and other modifications.
>
> Please let us know whether you have any further concerns or suggestions to improve this work’s quality.
>
> Thank you!

---

### Author Response · Authors · 2022-08-01
**General Response**

We thank all the reviewers for their time and constructive reviews. We are encouraged that the reviewers appreciate our first significant attempt toward the unsupervised multi-view image segmentation task for NeRF. We first address common concerns, followed by detailed responses to individual reviewers.

### Explanation for Eqn. 4 and 5 (Weakness #1 from Reviewer GXm6 and Weakness #1 from Reviewer QA4f)
We compute the label assignment at an arbitrary coordinate using Eqn. 4. However, the argmax operation in Eqn. 4 is not differentiable. An undifferentiable function here is undesirable because it prohibits backpropagation of the relevant gradient to the pre-softmax semantic logits $\mathbf{s}(\mathbf{x})$, which makes infeasible the optimization of the MLP representing the semantic field.

Therefore, we use $\hat{\mathbf{l}}({\mathbf{x}})$ in Eqn. 5 instead of $l({\mathbf{x}})$ in  subsequent computation. Notably, $\hat{\mathbf{l}}({\mathbf{x}})$ carries the same information as the assignment in Eqn. 4,  which is essentially its one-hot representation.
The term $\mathbf{s}(\mathbf{x})-\texttt{sg}(\mathbf{s}(\mathbf{x}))$ equals to 0 in value but has the same gradient as $\mathbf{s}(\mathbf{x})$ thus allowing  optimization of the MLP representing the semantic field.

### More complex scenes and failure cases (Weakness \#2, 3 and Problem \#1 from Reviewer GXm6 and Weakness from Reviewer 1Do1)

This first approach may have some difficulties in handling objects with rich textures or with a similar appearance to backgrounds. In our supplemental material, we show a failure case. So we give additional two failure cases in this [anonymous link](https://anonymous.4open.science/r/NeurIPS-Rebuttal-Materials-PaperID3608/failure%20cases.pdf). The first case is from CO3D dataset where the object of interest (hydrant) is severely occluded. The second is from instant-ngp repository, where the foreground object (fox) is near a patterned wall (Reviewer 1Do1).  Thanks to reviewers for pointing out this limitation, which we have highlighted in the revised version. We also provide in this [anonymous link](https://anonymous.4open.science/r/NeurIPS-Rebuttal-Materials-PaperID3608/DTU%20results.pdf) qualitative results on DTU dataset where the object is textured (Reviewer GXm6).

### Revised submission
We have submitted a revised submission with suggested modifications, stressed limitations (Reviewer GXm6, 1Do1 and fm5S) and an additional illustration figure (Review GXm6), which is also uploaded to this [anonymous link](https://anonymous.4open.science/r/NeurIPS-Rebuttal-Materials-PaperID3608/Revised%20submission.pdf).

---

### Author Response · Authors · 2022-08-03
**On the anonymous link**

Dear reviewers,

All the materials we refer to in the rebuttal can be found in this [anonymous link](https://anonymous.4open.science/r/NeurIPS-Rebuttal-Materials-PaperID3608).

However, we have found that sometimes this link cannot be opened properly. If this happens, please leave a comment and we will provide a new link.

Thanks,

Anonymous Authors

---

### Author Response · Authors · 2022-08-09
**On the rebuttal materials and revision**

Dear Reviewers, ACs and SACs,

Since the server of Anonymous GitHub, which we use for sharing rebuttal materials, has broken down, we added all of the rebuttal materials in the revised supplemental materials, along with the original supplementary document and video.

We also note that the 9-page limit is still valid for the revised version, so we removed the new figure and section in the revised version. However, the version with the figure for illustration and limitation section is still available in the supplemental materials. We will put them in the final version.


Thanks,

Anonymous Authors

---

### Meta-Review · Area_Chair_8w4U · 2022-08-26

**Recommendation:** Accept
**Confidence:** Less certain

**Metareview:**

Reviewers are generally positive about the submission, and all recommend acceptance post rebuttal.  They appreciate the new formulation and the strong results.  The AC agrees and recommends acceptance.

**Award:**

No

---

### Decision · Program_Chairs · 2022-09-14

Accept